



Winter phytoplankton blooms in the offshore south Adriatic waters (1995-2012) regulated by
hydroclimatic events: Special emphasis on the exceptional bloom of 1995.
Mirna Batistić[a], Damir Viličić[b], Vedrana Kovačević[c], Nenad Jasprica[a], Héloise Lavigne[c],
Marina Carić[a], Rade Garić[a], Ana Car[a]
[a]Institute for Marine and Coastal Research, University of Dubrovnik, Kneza Damjana Jude
12, 20000 Dubrovnik, Croatia
[b]Division of Biology, Faculty of Science, University of Zagreb, Rooseveltov trg 6, 10000
Zagreb, Croatia
[c]Istituto Nazionale di Oceanografia e di Geofisica Sperimentale, Borgo Grotta Gigante 42/c,
34010 Sgonico, Trieste, Italy
Coresponding author: Mirna Batistić (mirna.batistic@unidu.hr)
**Abstract.** The characteristics and intensity of winter phytoplankton blooms in the open South
Adriatic (OSA) were investigated by combining data on abundance and satellite-derived
surface chlorophyll (1995-2012). Particular attention was paid to the different circulation
regimes in the Ionian Sea, namely the anticyclonic and cyclonic Northern Ionian Gyres (NIG),
both of which influence the physical and biochemical properties of the South Adriatic.
Relatively high winter production was evident during both. Contrary to nutrient-poor cyclonic
years, in nutrient-rich anticyclonic years, shallow vertical mixing is sufficient for enrichment
of euphotic layers and bloom development. Moreover, intense blooms have occurred under
certain hydroclimatic conditions: the East Mediterranean Transient (EMT), extreme winters,
and reversal years that switch between anticyclonic and cyclonic circulation. Winter
phytoplankton bloom in February 1995, with microphytoplankton abundance exceeding $10^5$
cells $L^{-1}$, was related to the phenomenon of EMT which produced dramatic changes in the
East Mediterranean basin-wide circulation patterns. Dominance of a microphytoplankton
species uncommonly encountered in the OSA may be related to strong inflow of Atlantic
Water (AW) into the Adriatic during EMT and anticyclonic circulation in the NIG.
Keywords: phytoplankton, winter bloom, East Mediterranean Transient, hydroclimatic
changes, , BiOS regimes, open South Adriatic, Adriatic Sea, Mediterranean Sea
**1 Introduction**





According to current knowledge, the south Adriatic (SA) is oligotrophic and characterized by considerable seasonal variability of upper-layer physical, chemical, and biological properties. Present understanding of the phytoplankton community of the open south Adriatic (OSA) is scant and based mainly on episodic sampling during the warmer parts of the year (Viličić, 1991; Viličić et al., 1994; Viličić et al., 1989; Viličić, 1998; Turchetto et al., 2000). According to these and recent (Cerino et al. 2012) studies, phytoplankton abundance and biomass typically are low and smaller size species dominate.

Some spring peaks in microphytoplankton nevertheless have been observed (Viličić et al., 1989; Cerino et al, 2012). Santoleri et al. (2003) highlighted year-to-year variability in spring blooms in the OSA using Sea WiFS satellite surface observations of chlorophyll *a* and modeling for the period 1998 – 2000. These variations were associated with meteorological conditions and open-ocean convection (Gačić et al. 2002).

Besides atmospheric seasonality, incursion of different water masses into the OSA contribute, directly or indirectly, to the variability of phytoplankton abundance and biomass. The Bimodal Oscillating System (BiOS) plays a role in this regard as it influences exchange between the Adriatic and Ionian Seas by driving decadal changes in the Northern Ionian Gyre (NIG). The anticyclonic phase features advection of less saline Ionian water − diluted by Atlantic Water (AW), while the cyclonic phase is characterized by saltier Levantine/Cretan Intermediate Waters (LIW/CIW) flowing into the Adriatic (Civitarese et al. 2010).

Advection of AW presumably is accompanied by an increase of nutrients from upwelling at the periphery of the anticyclonic NIG (Civitarese et al. 2010). Conversely, cyclonic circulation in the Ionian Sea favors inflow of nutrient-poor LIW/CIW. After entering through the Strait of Otranto, these saltier waters have the potential to produce dense water in the SA by winter convective mixing. This injects nutrients into the euphotic zone, thereby stimulating spring primary production in the otherwise oligotrophic OSA (Gačić et al., 2002).

Another example of the intrusion of different water masses occurred in the early 1990s when areas of dense water formation switched from the SA to the Cretan Sea (Roether et al. 2007). This event, known as the East Mediterranean Transient (EMT), caused an abrupt change in eastern Mediterranean circulation and dense water properties. A direct consequence for the SA was advection of nutrient-rich, lower oxygen, less salty mid-depth water that upwelled to intermediate depths (i.e., Klein et al., 2000). As such, it flowed into the Adriatic and uniquely strengthened AW inflow (Klein et al., 1999; Borzelli et al., 2009). Civitarese and Gačić (2001) and Civitarese et al. (2010) suggested for the EMT period a local increase of primary





production and autotroph biomass in the southern Adriatic and Ionian Sea. However, the lack
of appropriate biological and chemical observations pertinent to the EMT peak period does
not allow a proper quantification of the related changes in the Mediterranean Sea.
Winter in the OSA generally has been considered a non-productive season with no significant
phytoplankton activity. The first indication of OSA winter blooms was in February 2008
(Batistić et al., 2012). The present study extends those observations with phytoplankton,
physico-chemical, and meteorological data from February 1994 and 1995, years strongly
affected by EMT and AW flow into the South Adriatic. These data are compared with those
collected in the winters of 2007, 2008, and 2012 (Cerino et al. 2012; Batistić et al. 2012;
Najdek et al. 2014) in the same area and under the both circulation regimes. The goal of this
comparison is to illuminate key factors in the development of SA winter blooms. Finally,
satellite-derived surface chlorophyll data (1997 - 2012) are analyzed to provide some insight
into the frequency and timing of OSA winter blooms.
The hypothesis are:
a) winter blooms are characteristic for the OSA and could account for a large fraction of OSA
annual production and the inter-annual variability in phytoplankton abundance and biomass
b) the OSA is not exclusively oligotrophic, as concluded in earlier investigations
c) both circulation regimes (anticyclonic and cyclonic) and different water masses (with
different nutrient loads) flowing into the SA provide conditions for OSA winter blooms
d) specific hydroclimatic events are responsible for an extraordinary intensive bloom in the
OSA

## 2 Material and Methods


Field study was conducted at three stations (P-100, P-300, P-1000 indicating position over the
isobate) in the south Adriatic (Fig. 1) with R/V "Bios" on February 27, 1994 and February 21,

91    1995.

Temperature and salinity profiles (0 to 1000 m, averaged over 1-m intervals) were taken with
a CTD multi-parametric probe (Sea-Bird Electronics Inc., USA). Potential temperature and
potential density were referenced to 0 dbar. Water samples were taken with 5-L Niskin bottles
at 0, 5, 10, 20, 50, 75, 100, 200, 300, 400, 600, 800, and 1000 m. Dissolved oxygen was
determined by the Winkler method and oxygen saturation ($O_2/O_2'$) was calculated from the



solubility of oxygen in seawater as a function of temperature and salinity (Weiss, 1970;
UNESCO, 1973). Nutrients were analyzed by standard oceanographic methods (Strickland
and Parsons, 1972).
Phytoplankton were collected with 5-L Niskin bottles and preserved in 2% neutralized
formaldehyde solution. Abundance was determined with an inverted microscope (Olympus IX
71) equipped with phase contrast. Microphytoplankton cells (>20 μm, MICRO) were counted
at a magnification of 200x in 2–3 transects of the central chamber and 100x in transects along
the rest of the base-plate. Depending on abundance, samples of 50–100 ml were settled in an
Utermöhl chamber (Utermöhl, 1958). Nanophytoplankton cells (2–20 μm, NANO) were
counted in 30 randomly selected fields along the chamber bottom at 400x. Whenever possible,
taxa were identified to the level of species or genus using standard keys, monographs, and
taxonomic guides.
Daily chlorophyll *a* products, with a 4-km resolution, delivered by the Ocean Colour Climate
Change Initiative (OC_CCI) project version 1.0 (http://www.esa-oceancolour-cci.org/) were
used to provide insights into SA blooms after 1995. The OC_CCI dataset, which covers the
period September 1997 to July 2012, was created by band-shifting and bias-correcting MERIS
and MODIS data to match SeaWiFS data and by merging the datasets with a simple average.
Meteorological parameters, air temperature, and wind were extracted from monthly
climatological reports issued by the Croatian Meteorological and Hydrological Service
(DHMZ) for the Dubrovnik station.
To track different circulation regimes in the North Ionian Gyre (NIG), we used average
salinity values from 1993 to 2012 in the 200-800 m depth layer from the Medatlas database
(MEDAR Group, 2002). Temporal changes in temperature and salinity were fitted to a six-
degree polynomial curve to capture the interannual trend and to detect years with cyclonic and
anticyclonic circulation in the NIG, as in Civitarese et al. (2010). Year 2012 display both
circulation modes: cyclonic mode which started in 2011, in the second part of the 2012 (May)
unexpectedly reversed to anticyclonic (Gačić et al., 2014), Fig. 2.

**3 Results**



**3.1 Meteorological conditions in February 1994 and 1995**

Time series of air temperature, wind direction, and wind speed in February 1994 and 1995 (Figs. 3 and 4) show one cooling event: 14-16 February 1994. This was associated with a cold, dry NE wind -- known in the region as *bura* -- of gentle to moderate intensity. Wind speed did not exceed 10 m s$^{-1}$ for the whole month, but *bura* episodes were quite frequent. Only two such episodes were observed in February 1995 (4 and 18 Feb) and wind speed was greater than 10 m s$^{-1}$ in each case. There were no significant cooling events.

In the period from 1989 to 1999, 1994 was the warmest year recorded in the Southern Adriatic (Cardin and Gačić, 2003); 1995 was neither exceptionally warm nor cold. Within this 10-year period, heat loss from the sea to the atmosphere in February of both years was lower, and surface water was warmer, than average. February 1995, in fact, was even warmer than February 1994. Conditions in those months thus were unfavorable for convection, although sporadic wind-induced mixing was possible.

**3.2 Physico-chemical conditions in February 1994 and 1995**

**3.2.1 Temperature, salinity, density, and oxygen variations**

February 1994 featured relatively uniform temperatures (12.84 - 13.08°C), salinities (37.86 - 37.92), and densities (28.60 - 28.69) at nearshore station P-100 (Fig. 5). Offshore stations P-300 and P-1000 had inverse stratification with cooler (max. 12.76 °C) and fresher (37.95) water near the surface (0-50 m), as depicted in Fig. 5. Temperature and salinity increased between 50 and 300 m at P-300 (13.08 - 13.87°C and 38.04 - 38.68) and P-1000 (13.25 - 13.83°C and 38.24 - 38.63), and then decreased gradually below 300 m. Density was highest near the bottom: 29.09 at P-300, 29.16 at P-1000 (Fig. 5).

The water column was well aerated at all stations. Mean oxygen saturation was 1.01 at P-100 and 0.93 at both P-300 and P-1000 (Fig. 5).

Inverse stratification at P-100 in February 1995 (Fig. 6) was owed to surface cooling (12.09°C). Temperature increased with depth, reaching 13.79°C at 100 m. Salinity and density increased gradually with depth: 36.08 - 38.13 and 27.41 - 28.65, respectively.





Temperature was relatively uniform at P-300 (13.71-13.87°C), except at 150 m where water
was cooler (13.3°C, Fig. 6). Salinity and density increased with depth (38.32 - 38.69 and
28.80 - 29.08, respectively) and were highest below 150 m (Fig. 6).
Thermal conditions were rather uniform (13.33-13.45°C) from the surface to 400 m at P-1000
(Fig 6). Temperature decreased slightly below 400 m (min. 12.85°C). Salinity and density
profiles also were uniform: 38.52-38.61 and 29.04-29.16, respectively.
Oxygen saturation averaged 0.86, 0.77, and 0.72 at P-100, P300, and P-1000, respectively
(Fig. 6). Lower oxygen below 200 m -- and extremely low concentrations at 600 m -- indicate
reduced deep ventilation at P-1000. This dramatic decrease and increased Apparent Oxygen
Utilization in 1995 (Lipizer et al 2014) are a consequence of EMT, which caused intrusion of
less-saline, oxygen-poor "old" Eastern Mediterranean Deep Water (EMDW) into the Adriatic
between 300 and 600 m (Klein et al., 1999, Klein et al., 2000, Manca et al., 2003).
The two February cruises differ principally in the fresher upper layer confined to the eastern
coast at the shallow station, and outcropping of the 29.0 isopycnal in February 1995, while
less quantity of the fresh water was widepread toward the open sea in Februray 1994.

**3.2.2 Nutrients**
Nitrate and silicate were low and relatively homogeneous from the coast to the open sea in the
upper 75 m in February 1994 (Fig. 7). They increased markedly below 100 m (Fig. 7) at both
P-300 (max. 5.29 µM nitrate; 5.9 µM silicate) and P-1000 (max. 4.08 µM nitrate; 12.46 µM
silicate).
Ammonia increased offshore (Fig. 7). Vertical profiles at P-100 show an increase at 75 m
(0.31 µM); a relatively homogenous distribution at P-300 (0.18-0.27 µM); and a different
distribution at P-1000 (0.33-0.66 µM), with the maximum at 600 m. Phosphate had similar
patterns in the upper 100 m at all stations (Fig. 7). Between 200-300 m and 300-400 m at P-
300 and P-1000, respectively, phosphate increased slightly: 0.19 and 0.13 µM.
In February 1995, nitrate did not exceed 2.25 µM at either P-100 or P-300 at any depth, or
within the upper 50 m at deep station P-1000 (Fig. 8). Nitrate increased markedly below 50 m,
attaining its maximum, 4.77 µM, at 600 m.





Silicate had a different distribution at each station. Values were higher at the nearshore and
deep stations. The maxima were 8.24 µM at the surface of P-100, and 11.24 µM at 1000 m at
P-1000 (Fig. 8). P-300 had considerably lower values: max. 4.83 µM at 300 m.
Ammonia and phosphate distributions were different at each station (Fig. 8). Maxima were
0.66 µM ammonia at 150 m (P-300) and 0.26 µM phosphate at 5 m (P-100). Ammonia and
phosphate maxima (0.64 and 0.18 µM, respectively) were found at 600 m at P-1000.


**3.3 Nano- and microphytoplankton**
In February 1994, NANO abundance varied from $1.5 \times 10^4$ to $4.4 \times 10^5$ cells $L^{-1}$ and MICRO
abundance was about two orders of magnitude lower: $1.7 \times 10^2$ to $3.4 \times 10^3$ cells $L^{-1}$ (Fig. 9).
NANO abundances greater than $10^5$ cells $L^{-1}$ were found above 20 m.
MICRO had similar vertical distributions at P-100 and P-300, while values were higher below
20 m at P-1000. Abundances below 50 m also were higher, with the maximum ($3.3 \times 10^3$ cells
$L^{-1}$) at 200 m (Fig. 9).
Seventy-one taxa within 44 genera of MICRO were identified. These were mainly diatoms
(44 taxa), followed by dinoflagellates (19), coccolithophorids (5), two silicoflagellates, and
one prymnesiophyte (Table 1).
Unidentified flagellates (2-10 µm) were the major fraction (> 99 %) of NANO abundance.
and Among MICRO diatoms dominated (19 – 100 %, average 71%) along the transect (Fig.
10). Dinoflagellates contributed 1 to 55 % (average 21%) and generally increased below 20 m
at P-100 and P-300 (Fig. 10). They were most abundant ($10^3$ cells $L^{-1}$, or 55 % of the total) at
20 m at P-1000 owing to *Gymnodinium simplex*. Other groups (silicoflagellates and
coccolithophorids) made up less than 22 % of MICRO abundance.
The most abundant diatom (exceeding $10^3$ cells $L^{-1}$) was *Pseudo-nitzschia* spp. The more
frequent (in at least 65% of samples and with abundance $<10^3$ cells $L^{-1}$) were: *Coscinodiscus*
sp., *Nitzschia longissima*, and *Thalassionema nitzschioides*.












MICRO abundance was unusually high ($2.02 \times 10^5$ to $4.04 \times 10^5$ cells $L^{-1}$) in the upper 50 m of
P-1000 in February 1995 (Fig. 11). This exceeded or nearly equaled that of NANO in the
upper 5 m. NANO at P-1000 varied from $6.2 \times 10^4$ to $5.9 \times 10^5$ cells $L^{-1}$. Toward the coast,
MICRO in the upper 50 m was lower by two orders of magnitude with minimum of 1.17 x
$10^3$ cells $L^{-1}$ at P-100 (Fig. 11). MICRO abundance at P-300 increased with depth; the
maximum, $1.13 \times 10^5$ cells $L^{-1}$, was at 150 m.
Eighty-five species within 50 MICRO genera were identified. These were mainly diatoms (57
taxa), dinoflagellates (17), coccolithophorids (7), two silicoflagellates, and one each
prymnesiophyte and chyrsophyte (Table 1).
Within 2-10 μm cell size fraction of NANO, unidentified phytoflagellates contributed to more
than 79 % in terms of abundance. The coccolithophore *Emiliania huxleyi* was particularly
abundant in the 0-20 m layer ($1.8 \times 10^4$ - $3.2 \times 10^4$ cells $L^{-1}$) at P-300, contributing from 7 to
21% of abundance.
Diatoms dominated MICRO (57-100 %). The most abundant ($>2.0 \times 10^4$ cells $L^{-1}$) were:
*Asterionellopsis glacialis*, *Chaetoceros affinis*, *Ch. curvisetus*, *Ch. decipiens*, *Detonula*
*pumila*, *Lauderia annulata*, and *Lioloma pacificum.* The highest dinoflagellate contribution --
$5.2 \times 10^3$ cells $L^{-1}$ or 42% at 50 m at P300 -- was from *Prorocentum micans* (Fig. 12).

**3.4 Ocean color observations of surface chlorophyll concentration in the SA (1997-2012)**
Chlorophyll *a* along the SA transect (left panel) was the highest in winter (middle panel),
from December to March (right panel, Fig. 13). One winter period ranges from December
(year n-1) to March (year n). In December 1997, satellite Chl *a* data display high
concentrations. Higher levels were found in the winters of 1999, 2000, and 2012 (righ panel).
In those winters the highest Chl *a* concentration was observed almost during all winter
months. Lowest winter Chl *a* concentrations were observed in 2001, 2007 and 2011 (right
panel).

**4 Discussion**
Winter sampling in both 1994 and 1995 coincided with a large-scale change in thermohaline
circulation known as the East Mediterranean Transient (EMT) that drove nutrient-rich, lower
oxygen, less saline water to mid-depths of the Adriatic. This was accompanied by a massive



intrusion of Atlantic Water (AW). The result was a rapid change in the physical (Cardin et al.,
2011), biogeochemical (Civitarese et al., 2010; Vilibić et al., 2012), and biological properties
(Batistić et al., 2014) of the SA. The February 1995 bloom, with a microphytoplankton
(MICRO) maximum of 4.04 x$10^5$ cells L$^{-1}$, seems to be a consequence of these processes.
Nanophytoplankton (NANO) abundance was of the same order as MICRO in the upper 50 m.
This is not common for these oligotrophic waters dominated by small phytoplankton (Viličić
et al., 1989). The intensity of the winter 1995 bloom is comparable only to those of spring
(April) 1986 and 1987 (Viličić et el., 1989).
Phytoplankton blooms rely on nutrient availability coupled with high irradiation. In 1995,
advection of AW presumably was accompanied by an increase of nutrients from upwelling at
the periphery of the anticyclonic NIG (Civitarese et al., 2010). In adition, effect of the EMT
was intrusion of less saline, nutrient rich, oxygen poor "old" EMDW in intermediate and deep
layers of the South Adriatic, in addition to pure LIW (Manca et al., 2003). In general, due to
EMT, the changes in the vertical distribution of water masses in the Eastern Mediterranean
were associated with a significant upward nutrient transport which caused that nutrient-rich
water are more closer to the euphotic zone than previously (Klein et al. 1999). This reached a
peak in 1995 when, compared with other transient years, the largest concentration of nutrients
was observed with very shallow nutricline depths. In the eastern Ionian Sea as well as in the
South Adriatic nutricline depth was about 100 m (Klein et al., 1999, 2000). The high nutrient
levels supported a marked increase of phytoplankton in the OSA in February 1995. Nutrients
were reduced in the upper 50 m (likely owing to uptake) but were still high (>4.0 μM) below
50 m. In general, 1995 was a mild year and the low-salinity inflow that accompanies the
anticyclonic circulation in the NIG is responsible for greater vertical stability in the OSA
(Gačić et al., 2009). Under such conditions, when the intensity of winter vertical mixing is
weaker or even absent, nutrients from deeper layers are not available to phytoplankton.
Because of the relatively shallow depth of nutrient-rich water in February however, only very
shallow vertical mixing would have been sufficient to supply nutrients to the euphotic zone.
There were no cooling events in February 1995, but two episodes of a strong *bura* (about 11m
s$^{-1}$) might have been responsible for wind-induced shallow mixing. In addition, a frontal slope
(between coastal and open sea stations) was evident. Turbulent mixing at shelf-break frontal
zones between less dense coastal water and denser open sea water, or velocity shear along the
eastern coast and/or eddy instability are other mechanisms that can drive upper-layer mixing
favorable for primary production (Mann and Lazier, 2006). There thus was the possibility for





more than one bloom in winter 1995, with similar intensity as such as we recored on
February, 21. Unfortunately, SAT Chl *a* data for this year are not available to confirm this
assumption.
A clear increase in MICRO abundance was found at 150 m at station P-300 in February
1995. Similar accumulations (on the order of $10^5$ cells $L^{-1}$) of a similar species composition
were observed in surface layer (0-50 m) at P-1000. This peculiar characteristic is possibly a
consequence of the strong horizontal density gradient between P-1000 and P-300 as
evidenced by the isopycnal outcropping. The isopycnal slope in the upper layer suggests the
possibility that water sliding along isopycnals conveyed phytoplankton from the upper 50 m
at P-1000 to 150 m at P-300. This deep phytoplankton biomass could contribute to enriching
the water column with organic matter in winter and, with sufficient vertical mixing, any
viable phytoplankton might be raised to the more favorable surface light environment to seed
primary production.
Data presented herein support model results that predict increased production in the East
Mediterranean owing to circulation changes consistent with those associated with the EMT
(Stratford and Haines, 2002; Mattia et al., 2013). On the other hand, satellite Chl *a* data Sea
WiFS (D'Ortenzio et al., 2003) revealed no strong relationship between the EMT and
productivity in the Eastern Mediterranean. Those data, however, are from 1998 when the
EMT was in a waning phase (Borzelli et al., 2009).
Phytoplankton abundance in February 1994 was 2 - 3 orders of magnitude lower than in
February 1995. The water column was stratified and nutrients were very low in the upper 100
m, possibly owing to earlier phytoplankton growth. Nutrients could not be replenished from
deeper layers because of the lack of vertical mixing. According to the meteorological data,
there was one *bura*-related cooling event (mid-February) with gentle winds of about 4 m s$^{-1}$.
This apparently was not sufficient to reduce the buoyancy of the upper 300 m, thus there was
no vertical convection and enrichment of the euphotic layer. Neither was there evidence of a
frontal zone, as in February 1995. Zooplankton abundance, especially of copepods, however,
was high (480 ind/m$^3$) (Batistić et al 2003). Grazing is important in regulating diatom
abundance (Calbet 2001 and references therein), so this might be an additional indication of a
bloom that had matured earlier in winter and since had been cropped substantially by
zooplankton prior to our sampling.



The winter increase in phytoplankton abundance and biomass in the open South Adriatic was also evident in February 2007 and 2008, and in March 2012 (Table 2). The maximum phytoplankton abundance in February 2008 was at 400 m and tied to a vertical convective event (Batistić et al. 2012). MICRO abundance was an order of magnitude lower (Table 2) than in February 1995, perhaps because of the dispersal of cells throughout the water column by more effective mixing.

Anticyclonic circulation characterized the NIG in 1994, 1995, 2007, and 2008 (Gačić et al. 2010; Civitarese et al. 2010; Bessières et al., 2013). This drove the flow of less saline AW water into the Adriatic. However, OSA salinity was higher in February 2007 and 2008 than in February 1994 and 1995 (>38.70 *vs*. 38.60, Table 2). According to Mihanović et al., (2015) between 2006 and 2008 the BiOS reversal from cyclonic to anticyclonic was slow (2–3 years) indicating that the reversal did not completely change the Adriatic water mass properties in a short time as during the exceptional conditions of BiOS regime shift in the 1990s when the prevalence of low-salinity water masses in the Adriatic happened rapidly (in less than a year). Therefore, under the same circulation regime, several factors can regulate SA winter blooms. In 1995, the synergy of AW and EMDW strongly influenced by EMT, along with shallow mixing (Table 2), brought nutrients to the euphotic layer. This set conditions for more than one bloom. In 2007 and 2008 influence of nutrient rich AW were complemented by higher salinity water that enhanced deep vertical mixing and aditional enriches the upper layers with nutrients (Table 2).

The marked increase in OSA phytoplankton biomass at the end of March 2012 (Chl *a* up to 4.86 µg L$^{-1}$) occurred after intense cooling in February (Table 2). The extremely cold winter of 2012 led to formation of very dense water which enchanced deep vertical mixing (Najdek et al., 2014). Also, the ongoing cyclonic BiOS phase in the Northern Ionian Gyre unexpectedly switched to anticyclonic (Gačić et al. 2014).

Direct measurement of phytoplankton abundance and biomass in the open SA during winter period are limited, but Chl *a* concentrations derived from satellite images (December 1997 to 2012) provide useful information about longer-term trends. Combining both sources yields clear evidence of higher winter biomass over almost two decades.

Chl *a* concentrations from H-diagrams were higher in reversal years; that is, those in which circulation switches between cyclonic and anticyclonic modes (and *vice versa*). This includes 1997, 1998/1999, and those years affected by specific hydrographic conditions, such as 2012.





This supports the finding that BiOS reversals create conditions for effective upward transfer
of nutrients that stimulate phytoplankton growth (Civitarese et al., 2010).
H-diagram analysis also showed higher winter Chl *a* in cyclonic years: 2000, 2002, 2004 and
2006. In those years deep vertical mixing occured (Gačić et al., 2002, 2006; Manca et al.,
2003; Kovačević et al. 2003; Civitarese te al., 2010). But, in cyclonic years, because of deep
mixing and dispersal of cells throughout the water column, satellite imagery may
underestimate Chl *a*, as documented in February 2008 (Batistić et al. 2012).
The phytoplankton community displayed some noteworthy features in February 1995 at the
open-sea station (P-1000). As usual for winter in the OSA, diatoms dominated the MICRO
fraction (Cerino et al., 2012, Batistić et al., 2012) with high contributions of *Chaetoceros*. The
*Chaetoceros–Rhizosolenia* association is characteristic of the eastern Mediterranean (Kimor
1983, Kimor et al., 1987) and has been observed over the whole year in the Otranto Strait
(Viličić et al., 1995). *Chaetoceros* dominated winter 2008 (Batistić et al., 2012). Unusual for
the OSA, the winter diatom bloom of 1995 also included *Asterionellopsis glacialis, Detonula*
*pumila, Lauderia annulata,* and *Lioloma pacificum,* each of which exceeded $10^4$ cells L$^{-1}$. All
of these are associated with high nutrients and most thrive in upwelling and mixing
environments (Odebrecht *et al.* 1995; Cabeçadas *et al.* 1999; Rörig and Garcia 2003; Pannard
et al., 2008; Zúñiga et al., 2011, Ospina-Alvarez et al., 2014). *Lauderia annulata*, in
particular, grows fast under light-limitation (Sommer, 1994; Reigman *et al.* 1996). Viličić *et*
*al.* (1989) found *A. glacialis* in high abundances (>$10^4$ cells L$^{-1}$) in an SA spring bloom in
1987, but only on the western coast (near city of Bari).
These species are common and some populate blooms in the eutrophic waters of the northern
Adriatic coast (Fonda-Umani and Beran 2003; Bosak et al., 2009; Godrijan et al., 2013). The
diatom *A. glacialis* and cocolthophore *E huxleyi* provide there regular blooms, while *L*
*annulata* ocurs sporadicaly (Bernardi Aubri et al. 2004, Viličić et al. 2009). However, from
Novemeber to February 1995 *A. glacialis* and especially *L. annulata* were in a low abundance
at the station of the west part of the North Adriatic while *D. pumila* and *L. pacificum* have not
been recorded (R. Kraus, personal communication). This thus reduces possibility that
mentioned species are transported to the South Adriatic (SA) by West Adriatic Current.
Atlantic Water (AW) did, however, strongly influence the South Adriatic in 1995 (Civitarese
et al., 2010; Vilibić et al. 2012; Mihanović et al. 2015) and so could have been responsible for
transporting certain phytoplankton species. Moreover, in February 1995, new zooplankton



species for the Adriatic Sea (*Muggiaea atlantica*), common in the Atlantic Ocean and
Western Mediterranan, have been also recorded (Batistić et al. 2014).
Regarding AW's broader influence in the Mediterranean Basin in 1995, the early winter
diatom bloom in the Bay of Tunis (SW Mediterranean Sea) that year had salinity <36.8 and
was dominated by *A. glacialis* and *L. annulata* (Daly-Yahia Kéfi et al., 2005) at abundances
in excess of $10^4$ cells $L^{-1}$. The diatoms *A. glacialis* and *L. annulata*, and the coccolithophore
*E. huxleyi* were among the most abundant species in the upwelling of the Alboran Sea (SW
Mediterranean), an area strongly influenced by the Atlantic current (Mercado et al., 2005).
Further evidence of AW's influence in the OSA is the appearance of *D. pumila*, typical of the
Atlantic current (Lecal, 1957), at P-300 and P-1000 in the present study. *E. huxleyi* also
reached relatively high abundance ($>10^4$ cells $L^{-1}$) in the winter of 1995 in the OSA, This
species was characterized as a indicator of AW in the Ionian Sea (Rabitti, 1994).


**5 Conclusion**

Winter bloom are typical of the open South Adriatic Sea and can be sufficiently intense to
account for a significant fraction of the region's annual production. From this perspective, the
open SA is not exclusively oligotrophic.
The present data demonstrate that blooms can occur during both anticyclonic and cyclonic
phases of the Northern Ionian Gyre, but according to different mechanisms. During
anticyclonic years, the nutracline along the borders of the Ionian Sea is shallower. This favors
inflow of nutrient-rich water to the Adriatic in the 50-200 m layer, depths at which shallow
vertical mixing is sufficient to raise these essential nutrients to the euphotic zone.
The nutracline is deeper during cyclonic years and inflow to the Adriatic from the Ionian Sea
is poorer in nutrients. Deeper mixing thus is necessary to enrich the euphotic zone.
Intense blooms also occur under other circumstances: reversal years -- when cyclonic and
anticyclonic patterns switch back and forth; years influenced by the EMT; and when winter
conditions are particularly extreme. The substantial winter bloom of 1995 coincided with the
EMT, strong inflow of nutrient-rich AW into the Adriatic, and upwelling of nutrients from
intermediate layers. The dominant microphytoplankton of that bloom are not common in the
OSA and likely were introduced under the strong influence of AW that year.
This work highlights the importance of winter in the OSA production and concludes that it
must be considered explicitly in discussions of the OSA's biological oceanography. Winter





blooms intensity depends on different water masses that enter the South Adriatic conected

with BiOS mechanism, synergy of regional meteorology/climate variability  and mixing

processes that affect these water masses.

All of these occur on broader time and space scales than typically addressed in studies of

plankton dynamics, but they must be incorporated in comprehensive analyses of the area's

production ecology.

**6 Acknowledgements**

This work was supported by the Croatian Ministry of Science, Education, and Sports (Grant

Number 275-0000000-3186) and Croatian science foundation (AdMedPlan, IP-2014-09-

2945). Many thanks to Dr Nick Staresinic (Galveston, USA) for improving the language.

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






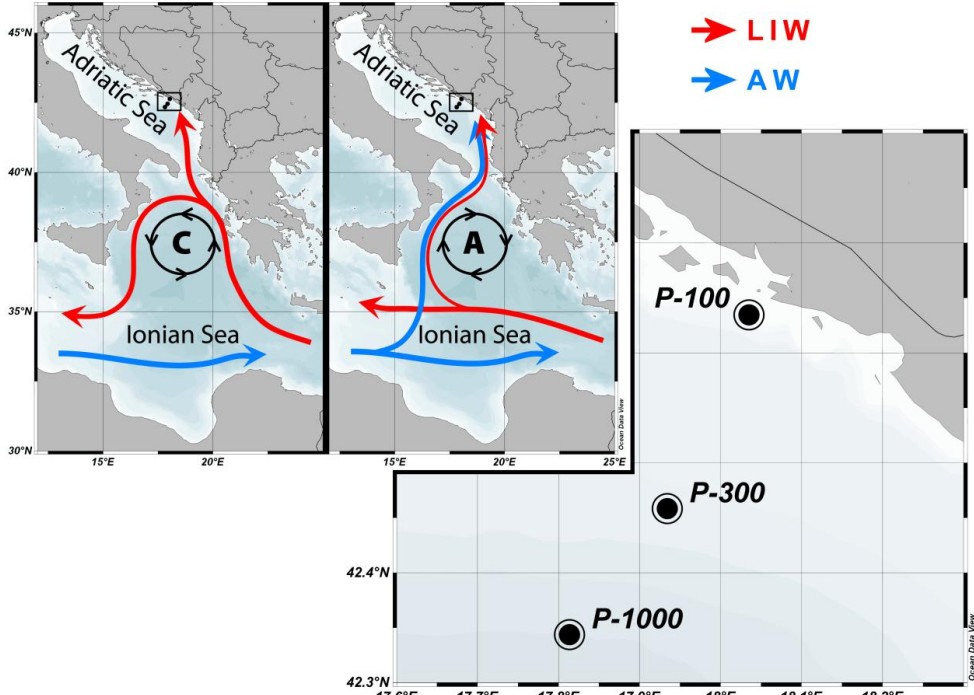

Fig. 1. Map of the southern Adriatic Sea indicating the study area and sampling stations and
different circulation patterns (A: anticyclonic, C: cyclonic) in the Ionian Sea, after Gačić et al.

650 (2010).






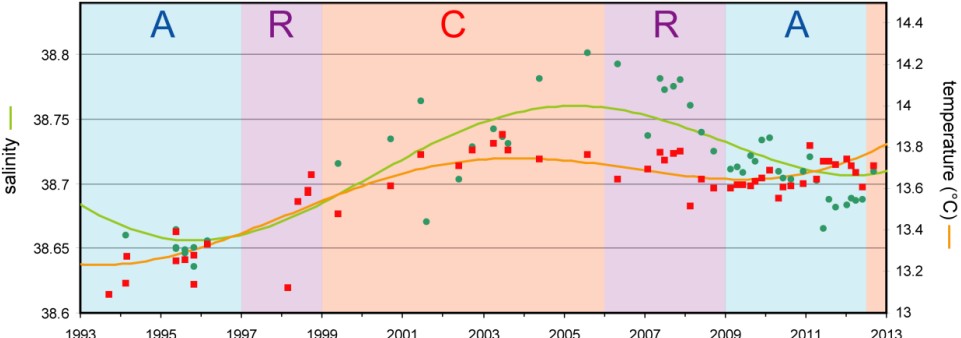

Fig. 2. Interannual variability of average temperature and salinity in the 200-800 m depth
layer in the South Adriatic from 1993-2012 indicating different circulation regimes in the
Northern Ionian gyre: anticyclonic (A) –blue, cyclonic (C) –red.  Reversal years (R) are in
purple and indicating period when NIG turned from anticyclonic to cyclonic and vice versa.
The trend was obtained by fitting a six-degree polynomial curve (after Civitarese et al., 2010).





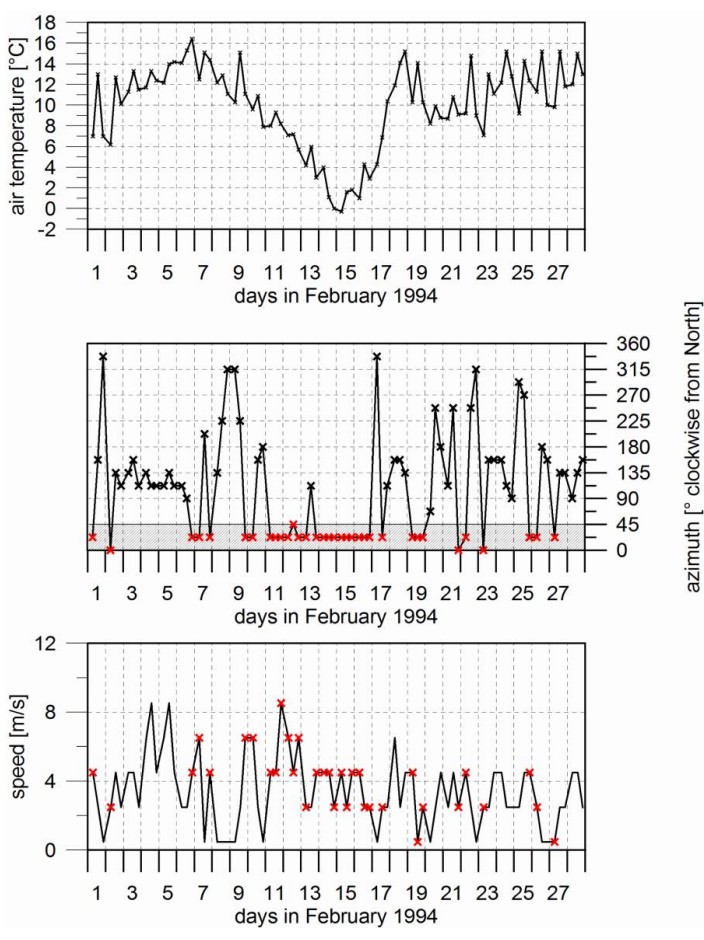


Fig. 3. Time series (1994) of air temperature, wind direction, and wind speed at the
Dubrovnik meteorological observatory (7, 14, 21 h). Red symbols correspond to *bura*
episodes. (directions between 0° and 45° azimuth, clockwise from north).





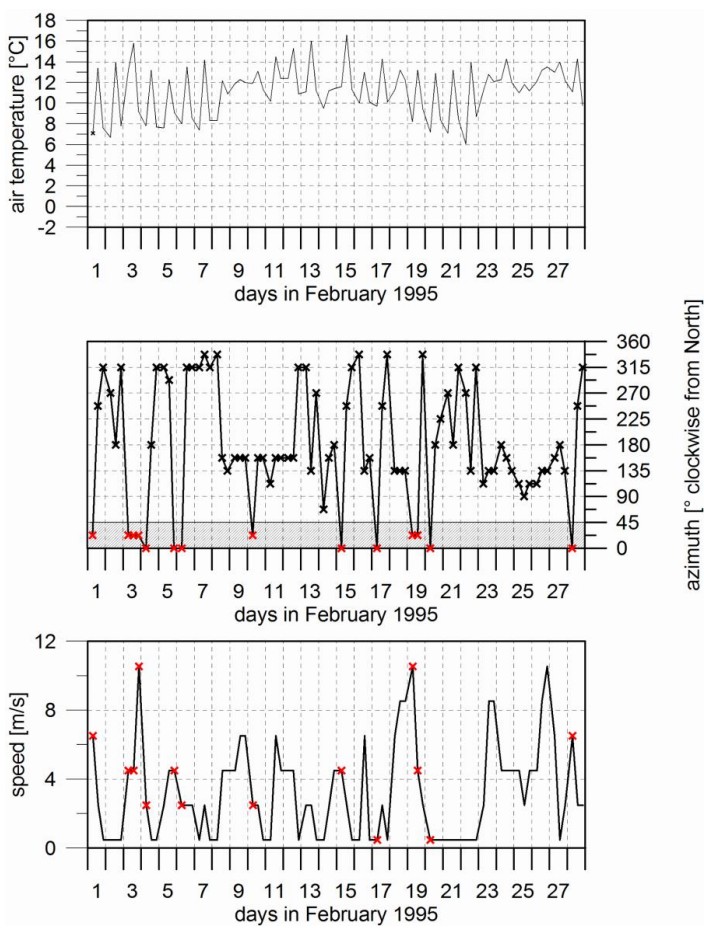


Fig. 4. Time series (1995) of air temperature, wind direction, and wind speed at the
Dubrovnik meteorological observatory (7, 14, 21 h). Red symbols correspond to *bura*
episodes. (directions between 0° and 45° azimuth, clockwise from north).





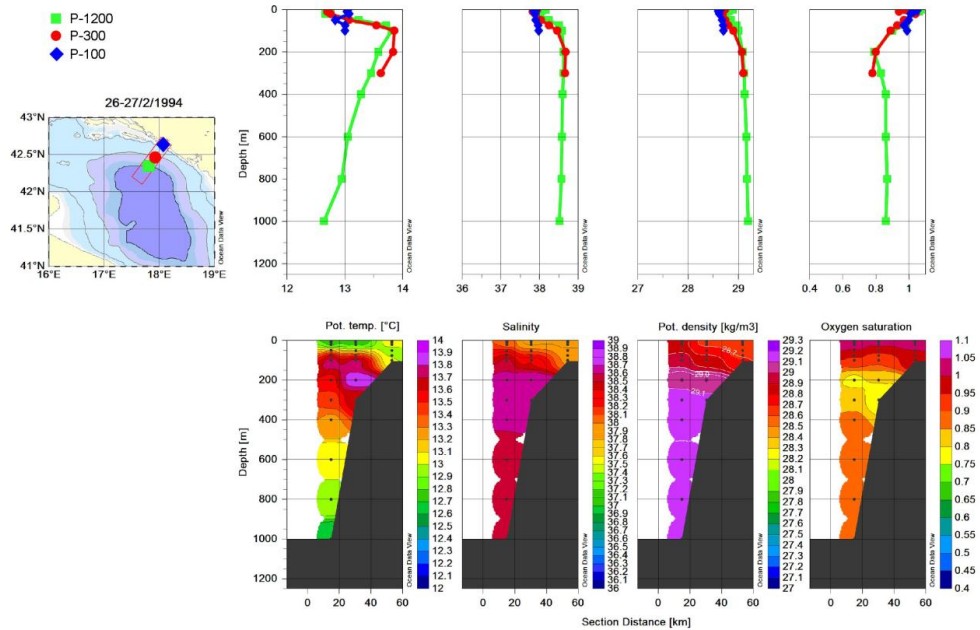


Fig. 5. Water properties in the study region in February 1994. Isopycnals 28.7, 28.8, 28.9, 29.0, 29.1 and 29.15 (white lines) are highlighted.






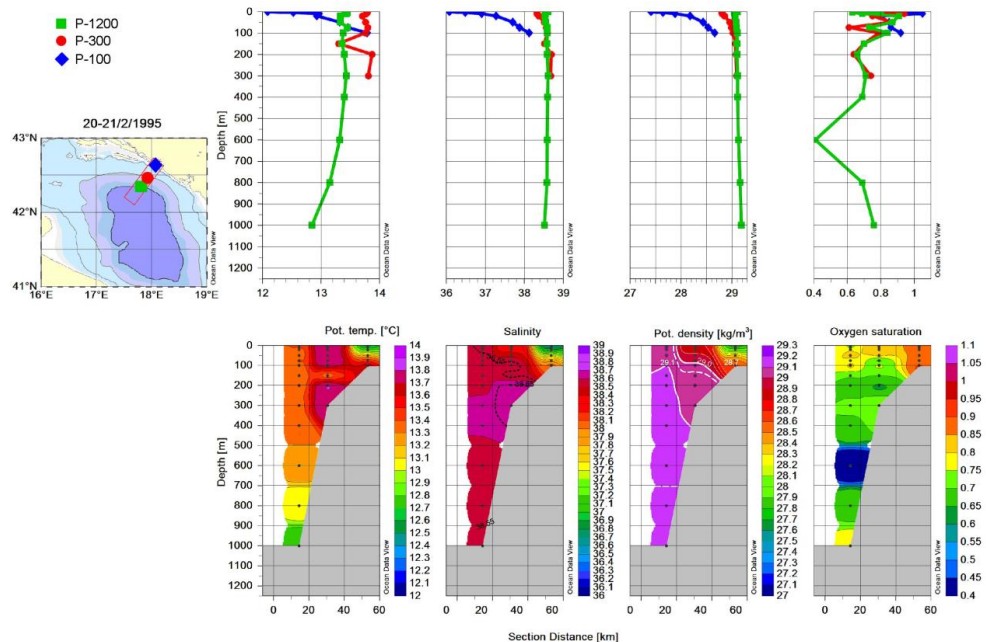


Fig. 6. Water properties in the study region in February 1995. Isopycnals 28.7, 28.8, 28.9,
29.0, 29.1 and 29.15 (white lines) are highlighted.





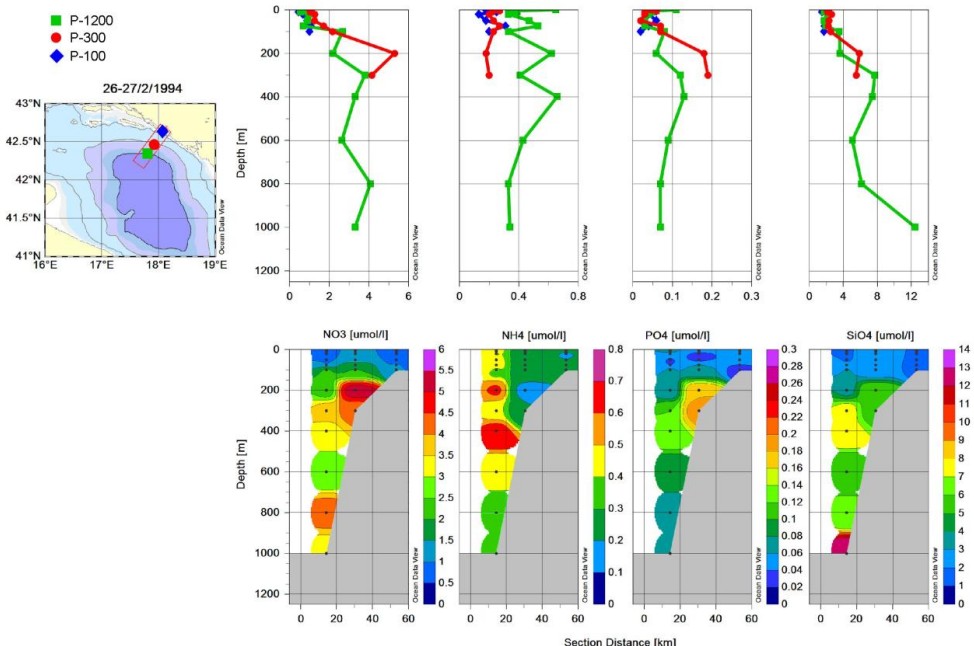


Fig. 7. Nutrient distributions, February 1994.

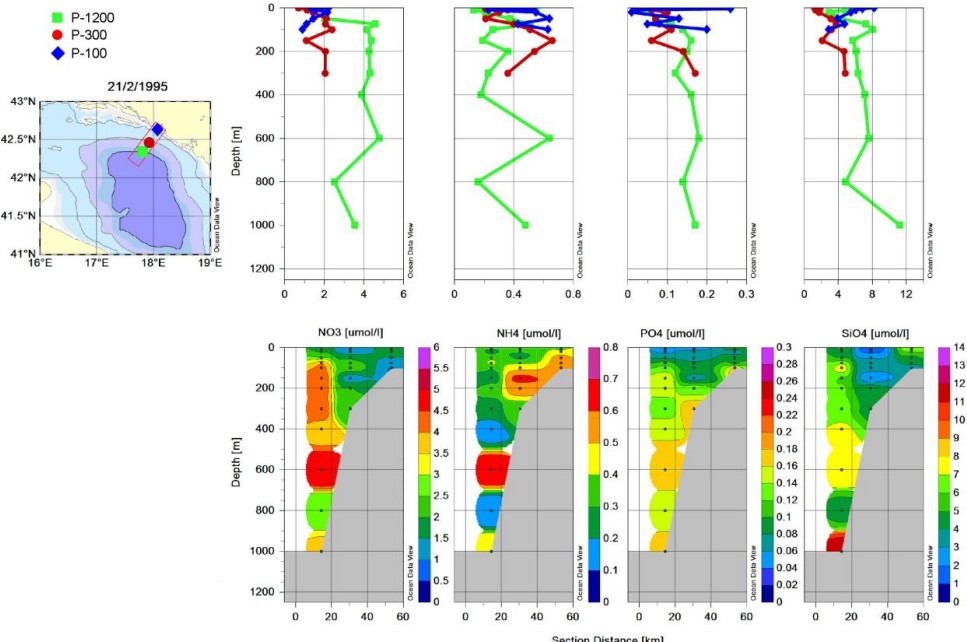


Fig. 8. Nutrient distributions, February 1995.






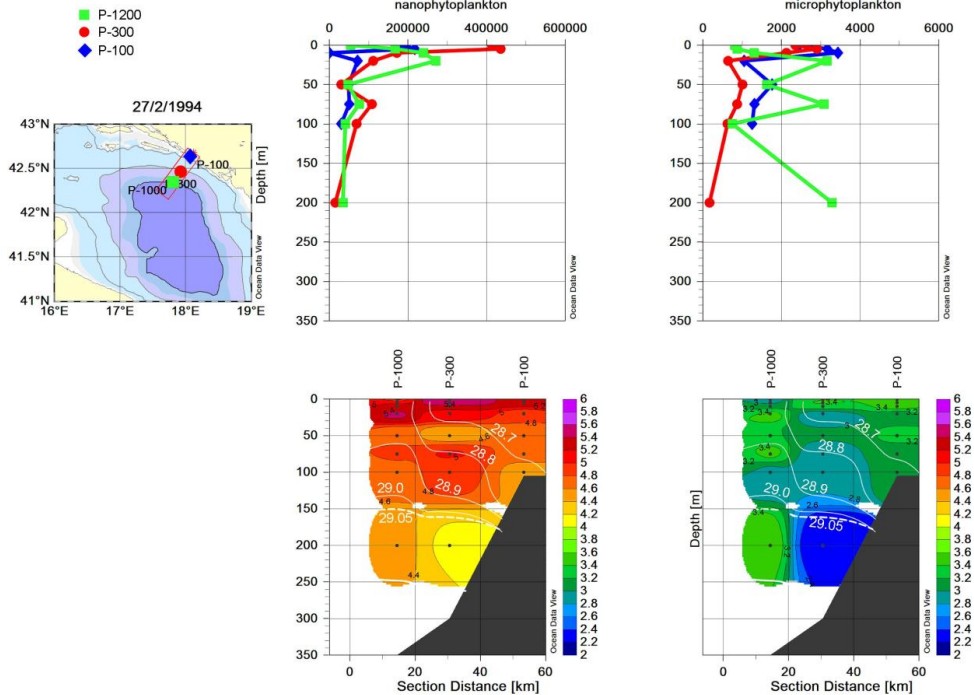


681 Fig. 9. Nano- and microphytoplankton distribution in February 1994. Vertical profiles (upper

682 panels, cells $L^{-1}$). Note: The scale for microphytoplankton is 100 times smaller than for

683 nanophytoplankton. The vertical distribution of abundance along the section (lower panel) is

684 on a log scale with superimposed isopycnal lines.



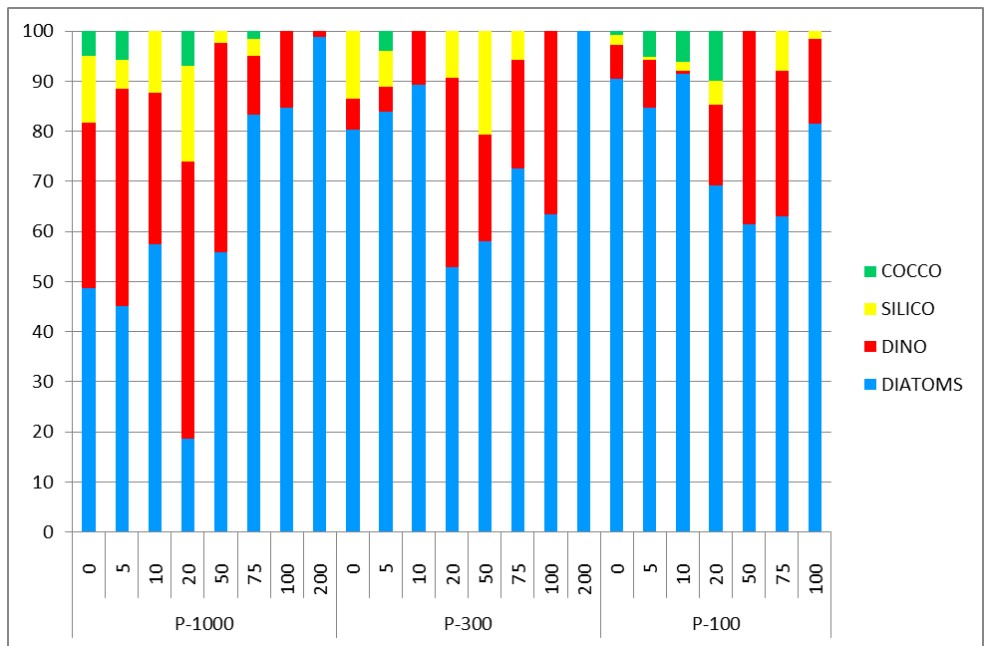

Figure 10. Percentage contribution of different taxonomic groups (COCCO = coccolithophorids, SILICO = silicoflagellates, DINO = dinoflagellates; and diatoms) to microphytoplankton abundance along the transect in February 1994.





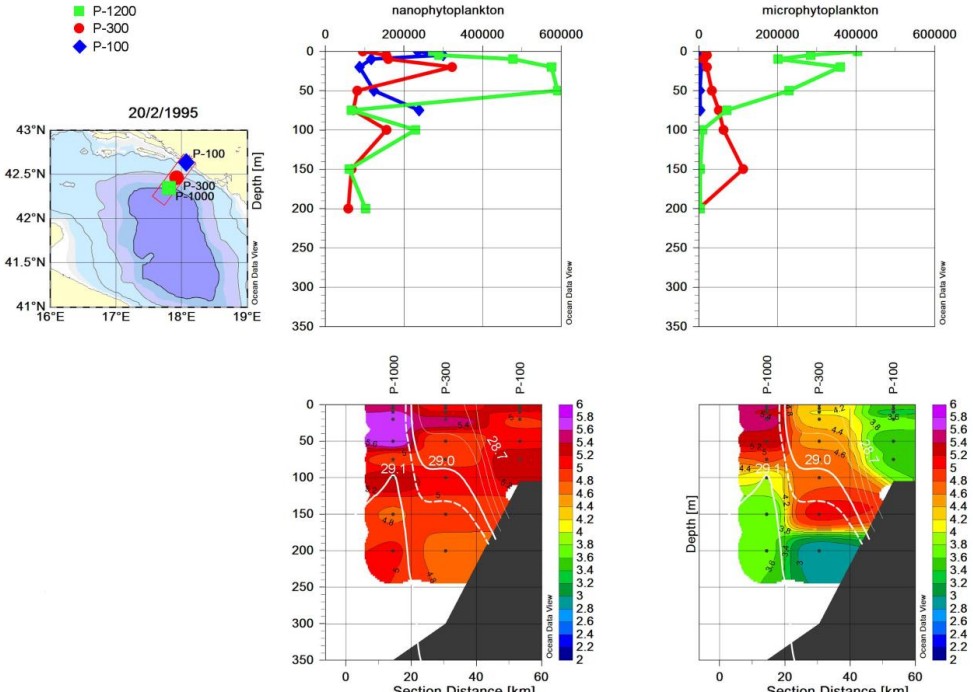


Fig. 11. Nano- and microphytoplankton abundance distribution in February 1995. Vertical
profiles (upper panels, cells L$^{-1}$). The vertical distribution of abundance along the section
(lower panel) is on a log scale with superimposed isopycnal lines.




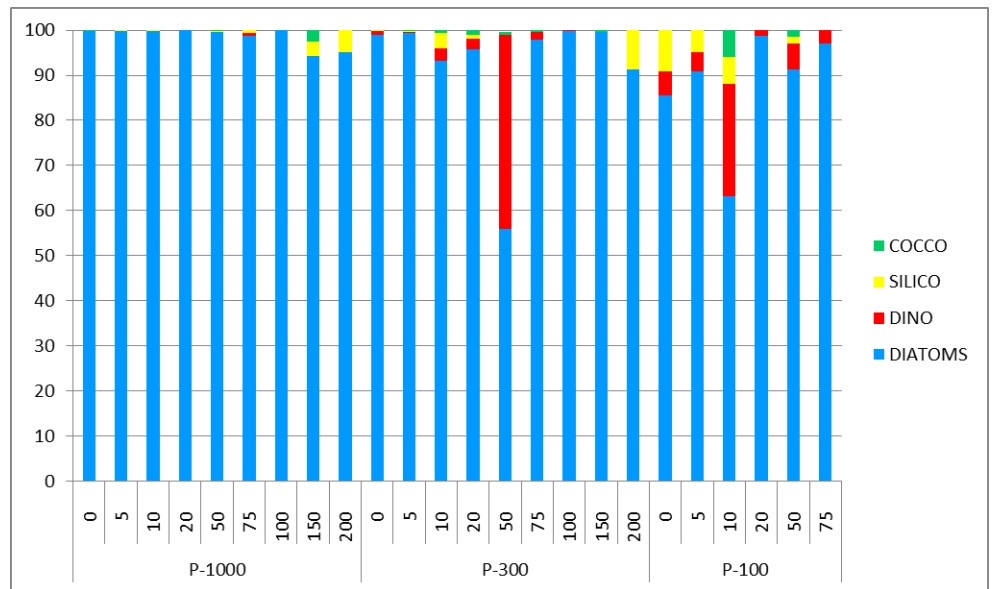

Figure 12. Percentage contribution of different taxonomic groups (COCCO = coccolithophorids, SILICO = silicoflagellates, DINO = dinoflagellates; and diatoms) to microphytoplankton abundance along the transect in February 1995.

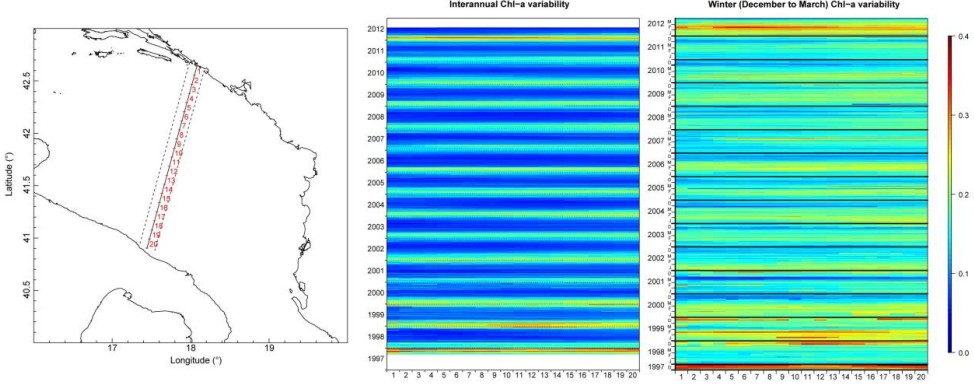

Fig. 13. Hoevmoeller diagram on South Adriatic. Figure on the left is geographical position of the transect, numbers indicates differents positions which are reported on the Hovmoeller diagrams (middle and right panels). The middle panel represents the full year Hoevmoller diagram whereas in the right panel only winter months are presented. Color is for chlorophyll *a* concentration in mg m$^{-3}$.
















Table 1. List of microphytoplankton taxa recorded in the South Adriatic in February 1994 and
719 1995.

| Years | 1994 | | | 1995 | | |
|---|---|---|---|---|---|---|
| Stations | P-1000 | P-300 | P-100 | P-1000 | P-300 | P-100 |
| Total number of taxa | 48 | 40 | 51 | 53 | 61 | 36 |
| **COCCOLITHOPHORIDS** | | | | | | |
| *Acanthoica quattrospina* Lohmann | . | . | . | + | + | . |
| *Anoplosolenia brasiliensis* (Lohmann) Deflandre | . | . | + | + | + | + |
| *Calciosolenia murrayi* Gran | + | . | . | . | . | . |
| *Coccolithus* sp. | . | . | . | + | . | . |
| *Ophiaster formosus* Gran | . | + | . | . | + | . |
| *Ophiaster hydroideus* (Lohmann) Lohmann | . | . | . | + | . | . |
| *Rhabdosphaera stylifera* Lohmann | + | . | + | . | + | . |
| *Syracosphaera pulchra* Lohmann | + | + | + | + | . | . |
| Unidentified coccolithophorids | + | . | + | . | + | . |
| **DIATOMS** | | | | | | |
| *Amphiprora sulcata* O'Meara | . | . | . | . | + | . |
| *Amphiprora* sp. | + | . | + | + | + | + |
| *Asterionella bleakeleyi* W.Smith | + | . | + | . | . | . |
| *Asterionellopsis glacialis* (Castracane) Round | + | + | + | + | + | + |
| *Asteromphalus heptactis* (Brébisson) Ralfs | . | . | . | . | + | + |
| *Bacteriastrum delicatulum* Cleve | . | + | + | + | + | . |
| *Bacteriastrum elongatum* Cleve | + | . | . | + | + | . |
| *Bacteriastrum mediterraneum* Pavillard | . | . | + | + | + | . |
| *Cerataulina pelagica* (Cleve) Hendey | . | + | + | + | + | + |
| *Chaetoceros affinis* Lauder | . | + | + | + | + | + |
| *Chaetoceros anastomosans* Grunow | . | . | . | + | + | . |
| *Chaetoceros brevis* Schutt | + | . | + | + | + | . |
| *Chaetoceros coarctatus* Lauder | . | . | . | + | . | . |




| | | | | | | |
|---|---|---|---|---|---|---|
| *Chaetoceros compressus* Lauder | . | + | + | + | + | + |
| *Chaetoceros convolutus* Castracane | + | . | + | + | + | + |
| *Chaetoceros curvisetus* Cleve | + | + | + | + | + | + |
| *Chaetoceros danicus* Cleve | + | + | + | + | + | . |
| *Chaetoceros decipiens* Cleve | + | + | + | + | + | + |
| *Chaetoceros didymus* Ehrenberg | + | . | . | + | . | . |
| *Chaetoceros diversus* Cleve | . | . | . | + | + | . |
| *Chaetoceros lauderi* Ralfs | . | . | . | + | . | . |
| *Chaetoceros lorenzianus* Grunow | . | . | . | + | . | . |
| *Chaetoceros simplex* Ostenfeld | . | . | . | + | + | . |
| *Chaetoceros tetrastichon* Cleve | . | . | . | . | + | . |
| *Chaetoceros tortissimus* Gran | . | . | . | . | . | + |
| *Chaetoceros vixvisibilis* Schiller | . | . | . | + | + | . |
| *Chaetoceros* sp. | + | . | + | + | + | . |
| *Corethron hystrix* Hensen | . | . | . | + | + | . |
| *Coscinodiscus* sp. | + | + | + | + | + | . |
| *Dactyliosolen fragilissimus* (Bergon) Hasle | + | + | + | + | + | . |
| *Detonula pumila* (Castracane) Gran | + | . | . | + | + | . |
| *Diploneis* sp. | . | + | + | . | + | . |
| *Eucampia cornuta* (Cleve) Grunow | . | + | + | . | . | + |
| *Grammatophora oceanica* Ehrenberg | + | . | . | . | . | . |
| *Grammatophora* sp. | + | . | . | + | + | . |
| *Guinardia delicatula* (Cleve) Hasle | . | . | . | + | . | . |
| *Guinardia flaccida* (Castracane) H. Peragallo | . | + | . | + | + | + |
| *Guinardia striata* (Stolterfoth) Hasle | + | + | . | + | + | . |
| *Hemiaulus hauckii* Grunow | . | . | . | . | + | . |
| *Hemiaulus sinensis* Greville | . | . | . | + | + | . |
| *Lauderia annulata* Cleve | + | . | . | + | + | . |
| *Leptocylindrus adriaticus* Schroder | . | + | + | . | . | . |
| *Leptocylindrus danicus* Cleve | . | . | . | . | + | + |
| *Leptocylindrus mediterraneus* (H. Peragallo) Hasle | + | . | + | . | . | . |
| *Leptocylindrus minimus* Gran | . | . | + | . | . | . |
| *Lioloma pacificum* (Cupp) Hasle | . | . | . | + | + | + |
| *Nitzschia longissima* (Brébisson) Ralfs | + | + | + | + | + | + |
| *Paralia sulcata* (Ehrenberg) Cleve | + | . | . | . | . | . |
| *Pleurosigma* sp. | + | + | + | + | + | + |
| *Proboscia alata* (Brightwell) Sundström | . | + | + | + | + | . |
| *Proboscia indica* (H.Peragallo) Hernández-Becerril | . | . | + | . | . | . |
| *Pseudo-nitzschia* spp. | + | + | + | + | + | + |
| *Pseudosolenia calcar-avis* (Schultze) Sundström | . | . | + | . | . | . |
| *Rhizosolenia hebetata* Bailey | . | . | + | . | . | . |
| *Rhizosolenia imbricata* Brightwell | + | + | + | + | . | . |
| *Skeletonema marinoi* Sarno *et* Zingone | . | + | + | + | + | . |
| *Synedra* sp. | + | + | + | . | + | . |
| *Synedra toxoneides* Castracane | . | . | . | . | . | . |
| *Thalassionema nitzschioides* (Grunow) Mereschkowsky | + | + | + | + | + | + |
| *Thalassiosira eccentrica* (Ehrenberg) Cleve | + | . | . | . | . | . |
| *Thalassiosira rotula* Meunier | . | + | . | . | . | . |
| *Thalassiosira* sp. | + | . | . | + | + | + |
| *Thalassiothrix longissima* Cleve *et* Grunow | + | . | . | + | + | . |
| Unidentified pennate diatoms | . | + | + | + | + | + |
| **DINOFLAGELLATES** | | | | | | |
| *Ceratium furca* (Ehrenberg) Claparède *et* Lachmann | + | + | . | . | . | . |
| *Ceratium fusus* (Ehrenberg) Dujardin | + | . | + | . | . | + |
| *Ceratium macroceros* (Ehrenberg) Vanhöffen | . | . | . | . | . | + |
| *Ceratium pentagonum* Gourret | . | . | + | . | . | + |





| | | | | | |
|---|---|---|---|---|---|
| *Ceratium symmetricum* Pavillard | + | . | . | . | . | . |
| *Ceratocorys gourretii* Paulsen | . | . | . | . | . | + |
| *Diplopsalis* group | . | . | . | . | + | . |
| *Gonyaulax* sp. | . | . | + | . | . | . |
| *Gymnodinium cf. simplex* | + | + | + | + | + | + |
| *Gymnodinium* sp. | . | + | + | . | . | . |
| *Gyrodinium* sp. | + | + | + | . | + | + |
| *Noctiluca scintillans* (Macartney) Kofoid *et* Swezy | + | . | . | . | . | . |
| *Oxytoxum scolopax* Stein | + | . | . | . | . | . |
| *Oxytoxum* sp. | . | . | . | . | + | . |
| *Prorocentrum micans* Ehrenberg | + | + | + | + | + | . |
| *Prorocentrum triestinum* J.Schiller | + | + | . | . | . | . |
| *Protoperidinium diabolus* (Cleve) Balech | . | + | . | . | . | . |
| *Protoperidinium divergens* (Ehrenberg) Balech | + | + | . | . | . | + |
| *Protoperidinium globulus* (F.Stein) Balech | . | . | + | . | . | . |
| *Protoperidinium oceanicum* (Vanhöffen) Balech | . | . | + | . | + | + |
| *Protoperidinium pallidum* (Ostenfeld) Balech | . | . | . | + | . | . |
| *Protoperidinium* sp. | . | . | . | . | + | . |
| *Protoperidinium steinii* (Jørgensen) Balech | . | + | . | + | . | . |
| *Protoperidinium tuba* (Schiller) Balech | . | + | . | . | . | + |
| *Scrippsiella* sp. | . | . | . | . | + | + |
| *Triadinium polyedricum* (Pouchet) Dodge | + | . | . | . | . | + |
| Unidentified dinoflagellates | + | . | + | . | + | + |
| **CHLOROPHYTES** | | | | | | |
| *Halosphaera viridis* Schmidt | . | . | + | . | . | + |
| **SILICOFLAGELLATES** | | | | | | |
| *Dictyocha fibula* Ehrenb. | + | + | + | + | + | + |
| *Dictyocha speculum* Ehrenberg | + | + | + | . | + | . |
| **CHRYSOPHYTES** | | | | | | |
| *Dinobryon* spp. | . | . | . | . | . | + |
| **INCERTAE SEDIS** | | | | | | |
| *Hermesinum adriaticum* O.Zacharias | . | . | . | . | + | . |
| *Ebria tripartita* (J.Schumann) Lemmermann | + | + | + | . | . | . |



Table. 2. Comparison of winter phytoplankton abundance and biomass under different
hydro-meteorological conditions in different years.

| Deep Sea P-1000/1200 | NIG circulation | Salinity, (water column AVG) | Weather during the month | Deep convection | Bloom |
|---|---|---|---|---|---|
| February 1994 | AC | 38.38 | Mild, one cooling event, *bura* wind ≤ 4.2 m s⁻¹. *(DHMZ) | Absent | None at date of sampling |
| February 1995 | AC | 38.57 | Mild, no cooling events, two *bura* episodes >10 m s⁻¹. *(DHMZ) | Shallow vertical mixing | Yes. MICRO order of magnitude (o.m) 10⁵ cells L⁻¹ |
| | | | | | |





| February 2007 (Cerino et al., 2012) | AC | 38.74 (0-100 m) and 38.73 (200-1000 m), Cerino et al., 2012; Cardin et. al., 2011) | Mild (Cardin et al., 2011) | Medium-deep convection (Cardin et al. 2011) | Yes. MICRO o.m. $10^4$ cells $L^{-1}$ |
|---|---|---|---|---|---|
| February 2008 (Batistić et al., 2012) | AC | 38.76 | Strong cooling event, *bura* wind >10 m s$^{-1}$ *(DHMZ) | Yes (600m depth) | Yes. MICRO o.m. $10^4$ cells $L^{-1}$; satelite Chl *a* around 0.5 µgL$^{-1}$ |
| March 2012 (Najdek et al.,2014) | C (AC) | 38.70 | Extreme winter cold, long-lasting strong cooling with *bura* wind in February 2012 (Mihanović et al., 2013). March mild, no cooling events, two *bura* episodes of 8-9 m s$^{-1}$. *(DHMZ). | Yes (600 m depth) | 2.0 µg L$^{-1}$ Chl *a* (AVG, 0- 50 m). 4.86 µg L$^{-1}$ Chl *a* (max., 35 m) |

NIG circulation: AC = anticyclonic, C = cyclonic; *DHMZ = Croatian Meteorological and Hydrological Service;













