# Peer review of "Winter phytoplankton blooms in the offshore south Adriatic waters (1995-2012) regulated by"

_Biogeosciences, 2017_

## Referee Comment (RC1) · Anonymous Referee #1 · 26 Jul 2017

GENERAL COMMENT: The manuscript bg-2017-205 combines physical and biological oceanography aiming at describing the evolution of winter phytoplankton blooms in open waters of the Southern Adriatic. The topic is of great scientific interest as it tackles mechanisms for the development of phytoplankton bloom in an area generally considered as oligotrophic. This can unveil a different perception of the production of the southern Adriatic pelagic ecosystem under specific hydrologic and meteorological conditions. In doing this, authors relied on extensive scientific literature, which describes circulation regimes of the Northern Ionian Sea and Southern Adriatic, including

regular circulation patterns and exceptional changes to these regimes over a 20-year long period (1993-2012, see Fig. 2). Authors' original contribution to the manuscript is the detailed description of winter situation of two years, 1994 and 1995, which were under the same circulation regime - the anticyclonic phase of the NIG. Another original contribution is the satellite Chlorophyll-a time series (1997-2012) along the E-W transect of the Southern Adriatic. The time-series was discussed in view of specific hydrologic and climatic conditions, which in certain years led to the development of a winter bloom, and were compared to the situation of winters 1994 and 1995 (Table 2). Again, information of comparable years were retrieved from literature. Saying this, in my opinion the title does not reflect the content of the paper properly. Firstly, the period mentioned in the title (1995-2012) does not match the sampling period (starts in 1994) nor the satellite time-series (starts in 1997). Secondly, both sampled years were assessed in the same comprehensive way, which does not justify emphasizing 1995 only. I would suggest modifying the title.

To summarize, authors' effort in describing complex hydrologic and climatic mechanisms that govern on large spatial and time scales in the Mediterranean and associate them to signals of change in phytoplankton community that occur on mesoscales in the Southern Adriatic is valuable. Moreover, observed differences in species composition in both years were supported by wide phytoplankton literature regarding the prevalence of certain species of Atlantic origin. I suggest minor changes of the manuscript on specific points, which need to be elaborated more in detail or re-interpreted.

SPECIFIC COMMENTS: 1. Material and methods (p. 3, lines 94-95): discrete sampling depths are listed from surface to the bottom of sampling stations. However, at these depths chemical parameters (oxygen, nutrients) were sampled, whereas is not specified that for phytoplankton community structure only the euphotic layer was sampled. (i.e. down to 200 m, as in the figures 9 and 10). This should be added. 2. Material and methods (p. 4, lines 109-113): provide information on the depth of surface layer of the Ocean Colour observations (i.e. surface Chl-a). 3. Material and methods (p.

[Figure]

4, lines 117-119) and Fig. 2: "To track different circulation regimes in the North Ionian Gyre (NIG), we used average salinity values from 1993 to 2012 in the 200-800 m depth layer..." It is not clear whether the average salinity was calculated for the upper 200-800 m deep layer OR for the layer at the depth of 200-800 m. If the latter is true then it contradicts the statement in Conclusions (p. 13, lines396-400) saying that during anticyclonic years the inflow into the Adriatic Sae can be observed in the 50-200 m layer. 4. Material and methods (p. 4, lines 121-123): check the statement "Year 2012 display both 121 circulation modes: cyclonic mode which started in 2011, in the second part of the 2012 (May) 122 unexpectedly reversed to anticyclonic (Gačić et al., 2014), Fig. 2". To me it looks just opposite; year 2011 and the first half of 2012 was in the anticyclonic mode, which in mid-2012 changed to cyclonic. 5. Results, subchapter 3.1 (p.5, lines 137-138): for non-physical oceanographers, explain more in detail (or rephrase the sentence) which conditions are unfavourable for convection. 6. Results, title of the subchapter 3.2. (p. 5, l. 140): I'd suggest changing it to "Physical and chemical properties of seawater in February 1994 and 1995" 7. Discussion (p. 8, lines 246-249): You claim that winters of 1994 and 1995 were characterised by the EMT "... that drove that drove nutrient-rich, lower oxygen, less saline water to mid-depths of the Adriatic. This was accompanied by a massive intrusion of Atlantic Water (AW)". Decreased oxygen at mid-depths was observed only in 1995, whereas salinity profiles of both years show a constant increase throughout the water column. Regarding nutrients, peaks were registered at different depths in both years: roughly from 200 to 400 m in Feb 1994 and around 600 m in Feb 1995. Are all these peaks related to the intrusion of EMDW and at which depth this water enters the Southern Adriatic? Authors should also mention which are the characteristics (salinity) of the AW and at which depth can be traced. 8. Discussion (p. 11, lines 320-321): "Anticyclonic circulation characterized the NIG in 1994, 1995, 2007, and 2008 (Gačić et al. 2010; Civitarese et al. 2010; Bessières et al., 2013)." According to Fig. 2, 2007 and 2008 were in the reversal phase. Moreover, do you have data, besides satellite observations of Chl-a, for years in the reversal phase - 1997, 1998 and 1999? It would be noteworthy to include them in Table 2 in which you summarise complex hydrologic and meteorological conditions that supported phytoplankton blooms. In reversal years, 1997-1999 different conditions governed than those presented in Table 2 but they anyhow led to the increase in phytoplankton biomass. 9. Conclusion: discussion and conclusion fairly answer the proposed hypotheses (p. 3, lines: 79-86). Nevertheless, when you are saying that winter blooms in the OSA could account for a large fraction of OSA annual production have you any indication for this statement. Can you sustain this evidence with some publish data of the inter-annual variability of primary production that could match years of winter blooms?

TECHNICAL COMMENTS: 1. Figure 1: add legend for LIW and AW. 2. Figures 5, 6, 7, 8, 9, 11: station P-1200 presented in these figures does not match the deepest station in Fig. 1 (P-1000) nor the description in the text. Intervals between labels on the salinity scale (Figs. 5 and 6) are too small. The reading of the salinity scale is unclear, as are unclear isopycnal contours especially on Fig. 5. Use a different colour. 3. Figures 10, 12: change the caption as follows: "Relative contribution (in %) of different.….". 4. Be consistent with the term coccolithophorid(s). 5. Table 1: Check the validity of species names (e.g., genus Ceratium) in updated databases, preferably in Algaebase. 6. Table 2: add legend for NIG and AVG. 7. References: Civitarese and Gačić (2001) and Kovačević et al. (2003) are missing in the list of references. 8. Citation (Rabitti, 1994) (p. 13, l. 389) has to be changed in (Rabitti et al., 1994) 9. Thoroughly check the style of citations, as they are not uniformly written out.

---

## Referee Comment (RC2) · Anonymous Referee #2 · 7 Aug 2017

The aim of this paper is stated as to 'illuminate key factors in the development of south Adriatic winter blooms'. The manuscript mostly reports data collected in 1994 and 1995 at three survey stations in the Adriatic Sea. Additionally Sea Wifs chlorophyll data is included in Hoevmolle figures covering the period 1997- 2012. I cannot recommend a paper that reports routine collected data from more than 20 years ago in a high profile journal such as Biogeosciences. The data presentation is basic and superficial with combinations of simple profiles and contour plots of density and nutrients in Figures 5-9. One has to ask why has it taken so long for the authors to report this data. Most

of the figures are far too small and why are they imbedded in the text? Fig 13 scales are not legible. The discussion is long and detailed and well referenced but I cannot see that the limited amount of new data presented in this manuscript is worthy of such detailed interpretation.

Method details of nutrient analysis- it is not sufficient just to cite Strickland and Parsons 1972 page 4 line 98. Why was formaldehyde used to fix phytoplankton this is non standard Lugols iodine is the normal phytoplankton preservative. page 4 line 100. Met data is described very superficially and under the methods heading page 5 lines 133-128. Oxygen saturation data should be as a % not as a fraction of 1 page 6 line 160.

---

## Author Comment (AC1) · 8 Aug 2017

Answers on Referee#2 comments and sugestions: 1. The most important comment is that the data on phytoplanton abundances originated from 1994 and 1995. We agree that these data are 22 years old, but they have not been published before and naturally could not be explained before due to the lack of knowledge on key environmental changes. In February 1995 we catched very unusual (for the open South Adriatic) high phytoplankton abundances. Now, in the light of the new knowledge of the hydroclimatic changes from 90 's and Bimodal Oscillating System (BiOS) as feedback mechanism

between Ionian and Adriatic Sea, this event can be explained. Actually, winter bloom was happened at the same time when East Mediterranean Transient (EMT) was at its peak. Civitarese et al. (2010) suggested for the EMT period a local increase of primary production and autotroph biomass in the southern Adriatic and Ionian Sea. However, till now, the lack of appropriate biological and chemical observations pertinent to the EMT peak period did not allow a proper quantification of the related changes in the Mediterranean Sea. Additionally, in that time also satellite chlorophyll observation (SeaWiFS) did not exist to confirm high phytoplankton abundance during the wintertime. In general, the data on phytoplankton during winter months are very scarce for the open South Adriatic. One reason is that winter in the open South Adriatic generally has been considered a non-productive season with no significant phytoplankton abundances. Therefore, we used satellite data from the first available year 1997 and after till 2012 (presented in details with Hoevmoeller diagram, Fig. 13) in order to investigate if winter bloom in the open South Adriatic was happened only during the EMT or maybe occur regularly.

2. Discussion is long due to coupling of the physical-chemical and biological status of the South Adriatic in last two decades. This includes data collected in the fields, satellite Chl-a, and already published data both on phytoplankton and environment.

3. Methodology approach:

Preservatives: Phytoplankton samples were preserved with 2% neutralized formaldehyde solution, and analysed within one month after collection. A number of fixatives have been used in conjuction with the inverted microscope technique to enumerate phytoplankton, and no single fixation is ideal for all purposes. Formalin gave uneven results, as others preservatives, and this depends from different taxonomic group, size-fraction, dilution and storage-time, etc. (UNESCO 1976). Fixation and preservation varied even within the same genus (e.g. Chrysochromulina, sensu HÅLLFORS, G., MELVASALO, T., NIEMI, A. & VILJAMAA, H. 1979. Effect of different fixatives and preservatives on phytoplankton counts. Publications of the Water Research Institute,

National Board of Waters, Finland, No. 34., and references therein), evidently depending upon the species in question and external conditions (e.g. temperature, salinity, etc.). Some flagellate species were well preserved, some others not at all. As with Lugol, weakly silicified diatoms tended to dissolve with time. Considering the results using formaldehyde preserved samples (see Jasprica, 2000: Pelagic Ecology Methodology, and references therein), nearly 30% of the naked flagellates and monads 2-4 micrometers in size were lost, but no loss was recorded in >5 micrometers size-fraction. The smallest differences between the preservatives were obtained during the vernal diatom bloom (HÅLLFORS et al. 1979). The fairly heavily silicified diatoms predominating in this sample were rather indifferent to the various preservatives. Finally, the most recently papers dealing with the phytoplankton in the Adriatic Basin and in wider area using formaldehid as preservative are publishing, i.e. formalin is still in wide use (e.g., Bastianini et al. ,Medit. Mar. Sci., 17/3, 2016, 751-765; Stefano Accoroni, Patricia M. Glibert, Salvatore Pichierri, Tiziana Romagnoli, Mauro Marinic and Cecilia Totti, 2015, Harmful Algae 45 (2015); 14–25, Malešević et al. 2015 Acta Bot. Croat. 74 (2), 333–343, etc.).

Nutrients: The most common method for determination of nutrients is Strickland and Parsons (1972). This was cited in our paper, but we'll also add improved method for the ammonium determination (Ivančić and Degobbis 1984) which has also been applied in our analyses (Ivancic, I., and Degobbis, D. (1984). An optimal manual procedure for ammonia analysis in natural waters by the indophenol blue method. Water Res. 18, 1143–1147. doi: 10.1016/0043-1354(84)90230-6)

Oxygen saturation ($O_2/O_2'$): We can present oxygen saturation as %, but this parameter in literature has also common been presented as fraction (e.g., 87% = 0.87). Meteorology: Met data are well described and accordingly this, figures are very informative. The most important facts for the subject are highlighted (i.e. "There were no significant cooling events important for enhanced vertical convection."), but we can explain this in some more details.

Specific comments: We did not understand what "basic" data presentation mean. Moreover, phytoplankton abundance and environmental parameters are presented in figures (5-9, 11) with all details. However, we agree that some figures are small, and this will be improved. In Fig. 13, scale will be improved. We did not understand Rev#2 question "why figures imbedded in the text"? Actually, figures are not imbedded in the text, according to journal rules.

---

## Referee Comment (RC3) · Anonymous Referee #3 · 17 Aug 2017

The paper seeks to understand the factors governing winter phytoplankton blooms in the southern Adriatic over the period 1995 to 2012, and argues that the mechanisms of bloom formation are different in years of cyclonic and anti-cyclonic circulation.

The data set consists of detained measurements in the winters of 1994 and 1995 and a satellite chlorophyll transect for subsequent years.

The most interesting feature of the paper is the large bloom in February 1995, a range of sampling (CTD nutrients etc) confirms the intrusion of Atlantic water in the region

that has been published elsewhere, and was related to the observed bloom.

One might then expect, based on this initial observation, presentation of an annual time series of data that evaluated the magnitude and composition of the phytoplankton community at the study site, allowing a better quantitative understanding of the link between water circulation and phytoplankton biogeography in the region. However, this is completely lacking in the paper, with only the presentation of a T/S time series in Fig 2 and the chlorophyll transect of Fig 13 being available for the subsequent years. The discussion then relates chlorophyll concentration to oceanographic conditions, but lacks the depth of analysis that would have been afforded by a more complete data set including phytoplankton and in situ measurements over multiple years.

In many cases, and in particular figure 13, the figures are poorly prepared and very difficult to read and interpret.

In summary the paper presents an interesting observation from over 20 years ago, but the subsequent data and its analysis is too superficial to make a compelling case for the causal relationships that the authors claim. I cannot therefore recommend publication.

---

## Short Comment (SC1) · 17 Aug 2017

This interesting paper provides an insight into the evolution of winter phytoplankton blooms in the open waters of the southern Adriatic. The authors explained the occurrences recorded in the past and put them in a context coherent with the recent findings of the very complex hydrodynamics of this area and the satellite imagery data. At that time it was very unlikely that such phenomena, based solely on discrete phytoplankton abundances, could be explained. Nowadays, with the further support of satellite data it is much easier to catch and validate such events and determine their eventual

regularity.

---

## Author Comment (AC2) · 22 Aug 2017

Answers on Referee#3 comments: The general observation of the Reviewer#3 is lack of the data of phytoplankton abundance and composition in the open South Adriatic. Yes, this is the truth, and one of the important reasons why we wanted to present these unique data, of the high winter phytoplankton abundance in the midlle of the 90's in the open South Adriatic (OSA), to scientific community. The second reason was that winter in the oligotrophic OSA generally has been considered a non-productive season with no significant phytoplankton abundances. So we tried to verify if this statement is true

or maybe high phytoplankton abundance and Chl-a are regular events during the winter in the open South Adriatic. Therefore, we used data on the phytoplankton abundance in winter of 1994, 1995 and all available already published data (our own and from other authors) from 2007 to 2012 which are presented in Table 2. To fullfill the gaps in the data of winter phytoplankton we used satelite Chla-a (which is widely accepted in scientific literature) for providing insights into phytoplankton biomass. These data are presented using Hoevmoller diagrams and cover period from 1997 to 2012, and show as follows: a) whole annual period, from which everybody can clearly see high values in the begining of the year (winter); and b) winter time in each year with detail presentation of Chl-a distribution in the OSA during winter months (Dcember – March). Reviewer's comments that Hoevmoller diagrams are too small, can be easily edited. But, this can not be the problem in reading images because it can be enlarged by magnifing on the screen and every detail can become visible. This can also be applied in hard copy edition. Figures of hydrographical, chemical and biological parameters are presented in two ways (linear and by curves) and if presentations in curves are not so clear, linear presentations are very simple and clear. Anyhow, we can improve mentioned figures. Many thanks to Reviewer's comments, but we would appreciate if she/he would be so kind and give us detail explanation what is superficial in our analyses of the presented data. So we can improve our paper. We hope that our work is not discredited because of the lack of in situ data in the area during the past times. We can not change this fact but in spite of that, our explanations of registered winter bloom events in the OSA by coupling of bilogical and physical processes give a new cognitions that could be check in the future by targeting in situ data collection.

---

## Author Comment (AC3) · 31 Aug 2017

Answers on Referee #1 comments and sugestions:

We greatly appreciate all reviewer's comments and suggestions which have been accepted in revised version of the manuscript. Please find our response letter below.

General comments:

Reviewer (R): Title does not reflect the content of the paper properly. Firstly, the period

mentioned in the title (1995-2012) does not match the sampling period (starts in1994) Secondly, both sampled years were assessed in the same comprehensive way, which does not justify emphasizing 1995 only. I would suggest modifying the title.

Answer (A): We thank to reviewer comments and we changed tittle: "Is phytoplankton winter bloom characteristic for open South Adriatic waters or it occurs only during specific hydroclimatic events?

Specific Comments: a) Material and methods 1. R: (p. 3, lines 94-95): discrete sampling depths are listed from surface to the bottom of sampling stations. However, at these depths chemical parameters (oxygen, nutrients) were sampled, whereas is not specified that for phytoplankton community structure only the euphotic layer was sampled. (i.e. down to 200 m, as in the figures 9 and 10). This should be added.

A: We thank to reviewer comment and we'll put the phytoplankton sampling depths in Material and Methods (0-200 m).

2. R: Provide information on the depth of surface layer of the Ocean Colour observations (i.e. surface Chl-a).

A: Satellite sensor only observes the surface layer of the ocean call the penetration depth. This depth is defined by Gordon and McCluney (1975) as "the depth above which 90% of the diffusely reflected irradiance originates". This depth is generally shallow and only reaches 10 meter depth when water is very clear. Gordon, H. R. and W. R. McCluney. "Estimation of the depth of sunlight penetration in the sea for remote sensing." Applied optics 14.2 (1975): 413-416.

3. R: To track different circulation regimes in the North Ionian Gyre (NIG), we used average salinity values from 1993 to 2012 in the 200-800 m depth layer" It is not clear whether the average salinity was calculated for the upper 200-800 m deep layer or for the layer at the depth of 200-800 m. If the latter is true then it contradicts the statement in Conclusions (p. 13, lines396-400) saying that during anticyclonic years the inflow

into the Adriatic Sae can be observed in the 50-200 m layer.

A: The reason for this is the following: This layer (200-800 m) does not receive directly the Atlantic water, but this layer integrates the conditions of the salt quantity, after the winter convection in the Southern Adriatic Pit. Namely, if there is no significant Atlantic water inflow, and more salty waters enter into the Adriatic, both as a surface Ionian and intermediate Levantine and/or Cretan waters, the winter convection will transfer more salt into the water column between 200 and 800 m. The final depth will depend on the intensity of the convection. In order to capture the signal authors (Civitarese et al. 2010) take this layer as a reference. On the contrary, when the uppermost layer (first 50-200 m depths) receive more Atlantic water, the winter convection will transfer this less salty water into deeper layers, and dilute the effects of the saltier intermediate layer.

R: Check the statement "Year 2012 displays both 121 circulation modes: cyclonic mode which started in 2011, in the second part of the 2012 (May) 122 unexpectedly reversed to anticyclonic (Gacic et al., 2014), Fig. 2". To me it looks just opposite; year 2011 and the first half of 2012 was in the anticyclonic mode, which in mid-2012 changed to cyclonic.

A: The statement is true, and altimetric maps showed that the last cyclonic (C) mode started in 2011 but unexpectedly in 2012 reversed to anticyclonic (A). This can be inferred from the monthly absolute dynamic topography maps in their paper in Fig. 5 (Gacic et al. 2014). Anyhow, this fast change (C to A) could not be presented at long time scale so we put explanation in the text (Material and Methods). Cyclonic mode starts again in autumn of 2012, so for better insight we'll move red strip closer to the end of year.

R: Explain more in detail (or rephrase the sentence) which conditions are unfavourable for convection.

A: This sentence will be rephrased as follows:

Conditions in those months were unfavorable for convection, i.e. there are no cooling events and hence no density increase, although sporadic wind-induced mixing was possible.

In general, the vertical convection (that is, overturning of the water column, when the surface density increases) is controlled mainly by the two essential factors: atmospheric, and the hydrographic conditions, and occurs usually in areas characterised by the cyclonic gyres. The atmosphere cools the sea surface, through the heat flux exchange at the air-sea interface. Cooling is a consequence of a heat release into the atmosphere, both because of the cold air temperatures, and because of the evaporation. Both these processes are favoured by the cold and dry bura winds (Bergamasco et al., 1999; Beg Paklar et al., 2001; Jeffries and Craig, 2007). These atmospheric conditions are changing from winter to winter (Cardin and Gacic, 2003). In some winters, the absence of strong bura events provokes less convection, or less deep convection. The role of the hydrographic conditions is to facilitate or not the vertical convection. In case of less salinity and mild weather conditions, vertical convection is reduce or even absent (This was already explained in details in Discussion, see rows 270-284 in original version).

R: I'd suggest changing it to "Physical and chemical properties of seawater in February 1994 and 1995".

A: We agree.

b) Discussion

R: You claim that winters of 1994 and 1995 were characterised by the EMT "that drove nutrient-rich, lower oxygen, less saline water to mid-depths of the Adriatic. This was accompanied by a massive intrusion of Atlantic Water (AW)". Decreased oxygen at mid-depths was observed only in 1995, whereas salinity profiles of both years show a constant increase throughout the water column. Regarding nutrients, peaks were registered at different depths in both years: roughly from 200 to 400 m in Feb 1994 and

around 600 m in Feb 1995. Are all these peaks related to the intrusion of EMDW and at which depth this water enters the Southern Adriatic? Authors should also mention which are the characteristics (salinity) of the AW and at which depth can be traced.

A: The nutrient peaks (200-400 m in 1994 and below 600 m in 1995) lay below the intermediate salinity maximum. Thus, are contained in the lower salinity intermediate waters (older low salinity EMDW waters uplifted from the newly produced Aegean Dense waters overflowing during the EMT) which inflow into the SAP across the Strait of Otranto (eastern side). AW has salinity between 38.0 and 38.20 in the upper 100 m (down from the surface). It can be traced at the 50 m depth.

R: Anticyclonic circulation characterized the NIG in 1994, 1995, 2007, and 2008 (Gacic et al. 2010; Civitarese et al. 2010; Bessières et al., 2013)." According to Fig. 2, 2007 and 2008 were in the reversal phase. Moreover, do you have data, besides satellite observations of Chl-a, for years in the reversal phase - 1997, 1998 and 1999?

A: This statement is valid. According to Gacic et al. (2010), Civitarese et al. (2010), Bessières et al. (2013) years of 2007 and 2008 are anticyclonic. But, we marked these years as reversal due to results of Mihanović et al. (2015) who concluded that between 2006 and 2008 BiOS reversal from cyclonic to anticyclonic was slow (2–3 years). Therefore, actually indicating that the Adriatic water mass properties did not completely change in a short time as during the exceptional conditions of BiOS regime shift in the 1990s when the prevalence of low-salinity water masses in the Adriatic happened rapidly (in less than a year). Unfortunatelly, winter phytoplankton data does not exist. This was the reason why we used Chl-a satelite data.

R: When you are saying that winter blooms in the OSA could account for a large fraction of OSA annual production have you any indication for this statement. Can you sustain this evidence with some publish data of the inter-annual variability of primary production that could match years of winter blooms?

A: We'll rephrase the sentence and ommitt term "production". Actually, we haven't

mentioned primary production. According to our results (phytoplankton blooms) we just wanted to highlight the importance of winter season for the pelagic ecosystem of open South Adriatic in the whole.

Technical comments: All technical comments and suggestions are accepted and corrections will be included in the final version of the manuscript. In figures 5, 6, 7, 8, 9, 11 the deepest station P-1000 is indicated as P-1200 and these were corrected.

---

## Author Comment (AC4) · 7 Sep 2017

According to the suggestions of the reviewers, we improved the figures (5,6,7,8,9,11) for a better understanding of the problem.

[Figure]

[Figure]

Fig. 5. Water properties in the study region in February 1994. Potential temperature (°C), salinity, potential density (kg/m³), and oxygen saturation: vertical profiles at each station (upper panels); vertical distribution along the transect connecting the three stations (lower panels).

**Fig. 1.**

Fig. 6. Water properties in the study region in February 1995. Potential temperature (°C), salinity, potential density (kg/m³), and oxygen saturation: vertical profiles at each station (upper panels); vertical distribution along the transect connecting the three stations (lower panels).

**Fig. 2.**

[Figure]

Fig. 7. Nutrient concentrations in February 1994: vertical profiles at each station (upper panels); vertical distribution along the transect connecting the three stations (lower panels).

**Fig. 3.**

[Figure]

Fig. 8. Nutrient concentrations in February 1995: vertical profiles at each station (upper panels); vertical distribution along the transect connecting the three stations (lower panels).

**Fig. 4.**

[Figure]

Fig. 9. Nano- and microphytoplankton distribution in February 1994. Vertical profiles at each station (upper panels, cells L$^{-1}$). Note: The scale for microphytoplankton is 100 times smaller than for nanophytoplankton. The vertical distribution of abundance along the section (lower panels) is on a log scale. Isopycnals 28.7, 28.8, 28.9, 29.0 and 29.1 (extracted from the potential density distribution in Fig. 5) overlay the abundance colour contouring.

**Fig. 5.**

[Figure]

Fig. 11. Nano- and microphytoplankton abundance in February 1995. Vertical profiles at each station (upper panels, cells L$^{-1}$). The vertical distribution of abundance along the section (lower panels) is on a log scale. Isopycnals 28.7, 28.8, 28.9, 29.0 and 29.1 (extracted from the potential density distribution in Fig. 6) overlay the abundance colour contouring.

**Fig. 6.**